# Cellular senescence mediates fibrotic pulmonary disease

Marissa J. Schafer[1,2], Thomas A. White[1], Koji Iijima[3], Andrew J. Haak[4], Giovanni Ligresti[4], Elizabeth J. Atkinson[5], Ann L. Oberg[5], Jodie Birch[6], Hanna Salmonowicz[6], Yi Zhu[1], Daniel L. Mazula[1], Robert W. Brooks[7], Heike Fuhrmann-Stroissnigg[7], Tamar Pirtskhalava[1], Y.S. Prakash[4,8], Tamara Tchkonia[1], Paul D. Robbins[7], Marie Christine Aubry[9], João F. Passos[6], James L. Kirkland[1,4,10], Daniel J. Tschumperlin[4], Hirohito Kita[3] & Nathan K. LeBrasseur[1,2,4]

Idiopathic pulmonary fibrosis (IPF) is a fatal disease characterized by interstitial remodelling, leading to compromised lung function. Cellular senescence markers are detectable within IPF lung tissue and senescent cell deletion rejuvenates pulmonary health in aged mice. Whether and how senescent cells regulate IPF or if their removal may be an efficacious intervention strategy is unknown. Here we demonstrate elevated abundance of senescence biomarkers in IPF lung, with p16 expression increasing with disease severity. We show that the secretome of senescent fibroblasts, which are selectively killed by a senolytic cocktail, dasatinib plus quercetin (DQ), is fibrogenic. Leveraging the bleomycin-injury IPF model, we demonstrate that early-intervention suicide-gene-mediated senescent cell ablation improves pulmonary function and physical health, although lung fibrosis is visibly unaltered. DQ treatment replicates benefits of transgenic clearance. Thus, our findings establish that fibrotic lung disease is mediated, in part, by senescent cells, which can be targeted to improve health and function.

[1] Robert and Arlene Kogod Center on Aging, Mayo Clinic College of Medicine, Mayo Clinic 200 First Street Southwest, Rochester, Minnesota 55905, USA. [2] Department of Physical Medicine and Rehabilitation, Mayo Clinic College of Medicine, Rochester, Minnesota 55905, USA. [3] Division of Allergic Diseases, Department of Internal Medicine, Mayo Clinic College of Medicine, Rochester, Minnesota 55905, USA. [4] Department of Physiology and Biomedical Engineering, Mayo Clinic College of Medicine, Rochester, Minnesota 55905, USA. [5] Division of Biomedical Statistics and Informatics, Department of Health Sciences Research, Mayo Clinic College of Medicine, Rochester, Minnesota 55905, USA. [6] Institute for Cell and Molecular Biosciences, Newcastle University Institute for Ageing Newcastle upon Tyne NE4 5PL, UK. [7] Department of Metabolism and Aging, The Scripps Research Institute, Jupiter, Florida 33458, USA. [8] Department of Anesthesiology, Mayo Clinic College of Medicine, Rochester, Minnesota 55905, USA. [9] Department of Laboratory Medicine and Pathology, Mayo Clinic College of Medicine, Rochester, Minnesota 55905, USA. [10] Department of Internal Medicine, Mayo Clinic College of Medicine, Rochester, Minnesota 55905, USA. Correspondence and requests for materials should be addressed to N.K.L. (email: lebrasseur.nathan@mayo.edu).

Fibrosis and wound healing are fundamentally intertwined processes, driven by a cascade of injury, inflammation, fibroblast proliferation and migration, and matrix deposition and remodelling[1]. Older organisms display reduced ability to heal wounds[2] and resolve fibrosis[3], leading to tissue scarring and irreparable organ damage. The origins of persistent injury response and repair signalling underlying fibrotic tissue destruction are poorly understood. This is particularly true of idiopathic pulmonary fibrosis (IPF), a quintessential disease of ageing with median diagnosis at 66 years and estimated survival of 3–4 years[4]. IPF symptoms, including chronic shortness of breath, cough, fatigue and weight loss, are progressive and lead to a dramatic truncation of healthspan and lifespan. This is due to destruction of lung parenchyma, which exhibits characteristic honeycombing and fibroblastic foci patterns[1,5]. Current IPF treatment regimens have limited efficacy[6,7]. Better defining the mechanisms responsible for chronic activation of profibrotic mechanisms and lung parenchymal destruction is essential for devising more effective therapies.

Cellular senescence is an evolutionarily conserved state of stable replicative arrest induced by pro-ageing stressors also implicated in IPF pathogenesis, including telomere attrition, oxidative stress, DNA damage and proteome instability. Damage accumulation stimulates the activity of cyclin-dependent kinase inhibitors p16[Ink4a] and/or p53-p21[Cip1/Waf1], which antagonize cyclin-dependent kinases to block cell cycle progression[8]. Through secretion of the senescence-associated secretory phenotype (SASP), a broad repertoire of cytokines, chemokines, matrix remodelling proteases and growth factors, senescent cells paracrinely promote proliferation and tissue deterioration[8]. Conversely, senescence is autonomously anti-proliferative, may be requisite for optimal cutaneous wound healing[9] and may restrict pathological liver fibrosis[10].

A growing body of evidence implicates accelerated mechanisms of ageing, including cellular senescence, in IPF pathogenesis[11]. Established senescence biomarkers, including p16, p21 and senescence-associated β-galactosidase activity (SA-β-gal), have been observed in both fibroblasts and epithelial cells in human IPF lung tissue[12,13], and human IPF cells show increased senescence propensity ex vivo[14]. In mice, intratracheal instillation of the chemotherapeutic agent bleomycin causes resolvable lung fibrosis that recapitulates key features of human IPF[15]. Bleomycin lung injury induces a molecular signature of senescence[16,17] and age-dependent accumulation of senescent myofibroblasts may impede fibrosis resolution following bleomycin exposure[3]. In contrast, overexpression of an enzyme responsible for production of the extracellular matrix component hyaluronan exacerbates bleomycin-induced injury[18], whereas its depletion appears to activate senescence, challenging a negative role for senescence in bleomycin-induced lung fibrosis[19].

Very recently, Hashimoto et al. provided crucial evidence that cellular senescence contributes to lung ageing and may be targeted for functional improvement[20]. Using a novel suicide-gene strategy, they discovered that elimination of naturally occurring senescent cells restores lung compliance, structure and elasticity in aged mice. An alternative approach to transgenic senescent cell clearance is senolytics[21,22], which may be a tractable treatment option for humans. Administration of the senolytic quercetin diminishes the proinflammatory phenotype of bleomycin-induced senescence in fibroblasts in vitro[23]. Similarly, delivery of rapamycin, a SASP inhibitor, attenuates pulmonary fibrosis and myofibroblast activation in vivo[24]. The culmination of prior reports suggests that senescent cells may contribute to fibrotic lung disease; however, their mechanism of action and whether they may be therapeutically targeted to improve lung function remains untested.

Experimentation described herein explored the role of cellular senescence in IPF pathology across the translational continuum. We began by assessing human IPF and control biospecimens and demonstrate that several senescence biomarkers accumulate in IPF lung, with p16 expression increasing concordantly with disease severity. Using bleomycin-induced lung injury as an IPF model, we show that, similar to human IPF, murine lung fibrosis is characterized by accumulation of p16- and SASP-positive fibroblasts and epithelial cells. We hypothesized that SASP signalling is a mechanism by which senescent cells exert negative effects, and our in vitro experiments establish that the SASP of senescent fibroblasts is indeed fibrogenic. Critically, senescent fibroblasts are selectively eliminated through treatment with the senolytic drug cocktail, dasatinib plus quercetin (DQ). Next, we tested the efficacy of senescent cell deletion in improving bleomycin-induced lung pathology in Ink-Attac mice, in which p16-positive cells are deleted through suicide-gene activation. We show that senescent cell clearance improves pulmonary function, body composition and physical health when treatment is initiated at disease onset. Notably, senolytic DQ treatment phenocopies the transgenic cell clearance strategy. Thus, our results suggest that senescent cells, through their SASP, wield potent effects on adjacent cells, ultimately promoting functional lung deterioration. Our findings provide important proof-of-concept evidence for targeting senescent cells as a novel pharmacological approach for treatment of human IPF.

## Results

**Senescence biomarkers accumulate in IPF lung.** To explore the hypothesis that senescent cells and the SASP regulate lung fibrosis, we interrogated microarray and RNA sequencing (RNAseq) data sets corresponding to independent IPF and control human cohorts for differential expression of established senescence genes. IPF subjects exhibited significant impairments in lung function, as measured by forced vital capacity (FVC) and diffusion capacity, and physical function, as measured by the 12-item short form health survey physical component score and 6 min walking distance, relative to control subjects (Supplementary Tables 1 and 2). CDKN2A (p16) was significantly upregulated within lung samples of individuals with IPF and increased with disease severity (Fig. 1a). Correlation analyses revealed that elevated pulmonary p16 expression assessed via microarray was associated with reduced FVC, diffusion capacity and 12-item short form health survey physical component score (Supplementary Fig. 1).

To corroborate expression data, we investigated p16 cytospatial distribution using immunohistochemistry in a subset of control and IPF lung samples that were analysed by microarray. We identified a rare population of p16-positive epithelial cells in control lung samples (Fig. 1b). In IPF lung samples, both epithelial cells and fibroblasts were p16 positive within fibroblastic foci (Fig. 1c), the presumed leading edge of IPF disease. In the honeycomb lung, reactive bronchiolar epithelium and fibroblasts were equally positive for p16 (Fig. 1d). We next quantified an independent senescence biomarker, telomere-associated foci (TAF), which are sites of unresolved DNA damage within telomeres, demarcated by γH2A.X and telomere immuno-fluorescence in situ hybridization co-localization[25]. We observed a significant increase in both the mean number of γH2A.X foci and the percentage of TAF-positive cells in IPF samples, relative to controls (Fig. 1e,f). Cumulatively, senescence biomarker results demonstrate p16 expression increasing in register with disease progression, accumulation of p16-positive fibroblasts and

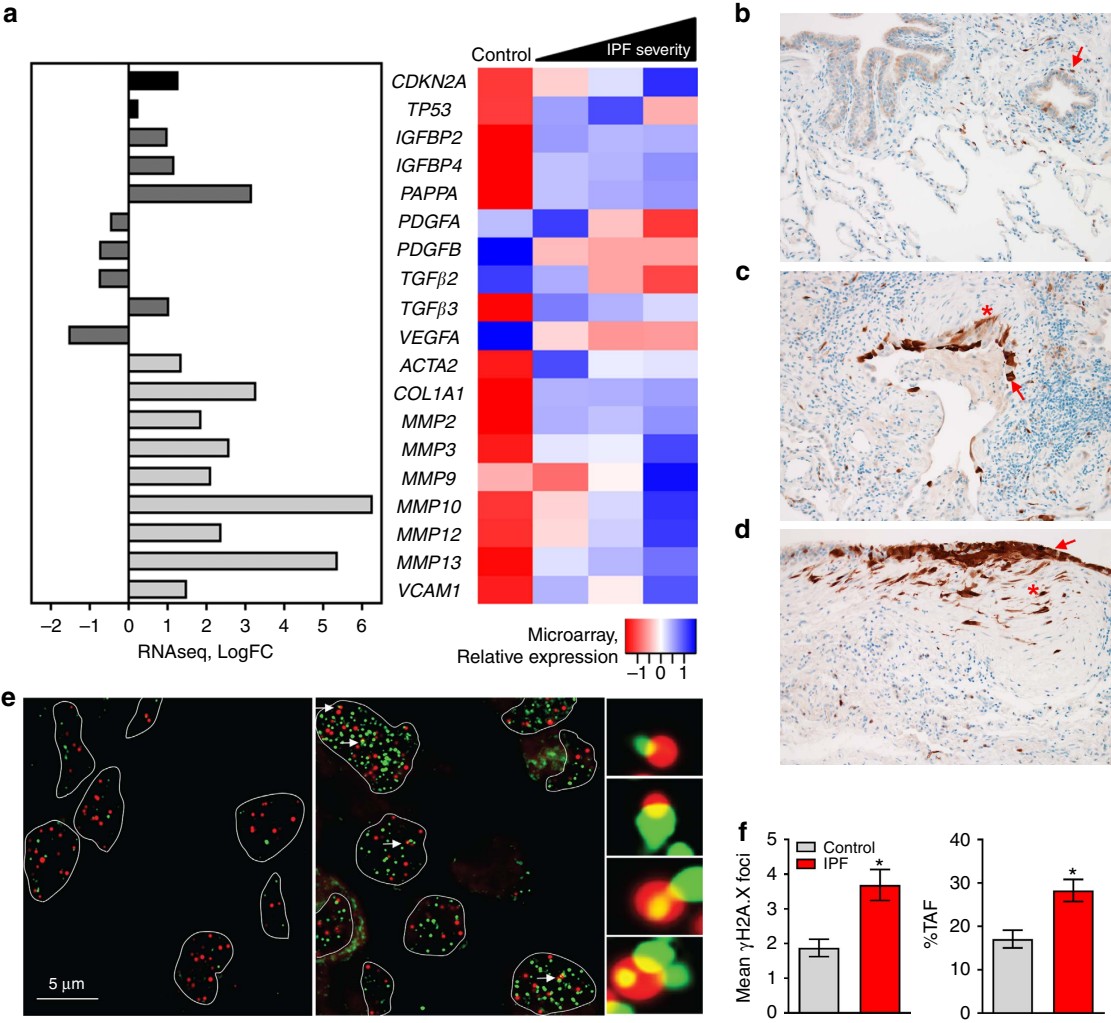

**Figure 1 | Biomarkers of cellular senescence in human IPF. (a)** Transcriptional changes corresponding to senescence effectors (black), SASP growth factors (dark grey) and SASP matrix remodelling (light grey) genes that were identified in independent RNAseq (control $n = 19$, IPF $n = 20$) and microarray human lung IPF versus control data sets are shown. IPF samples analysed by microarray were severity classified by FVC as low ($\geq 80\%$; $n = 17$), moderate (50–80%; $n = 60$) or severe ($<50\%$; $n = 16$) and compared with control ($n = 64$) ($q < 0.05$ for both RNAseq and microarray). Human lung tissue sections were IHC stained for p16 in **b** control and (**c,d**) IPF lung samples with (**c**) fibroblastic foci and (**d**) honeycomb lung depicted. p16-positive fibroblasts (stars) and epithelial cells (arrows) are indicated ($\times 200$ images). (**e**) Control (left panel) and IPF (right panel) lung sections were analysed for frequencies of DNA damage foci ($\gamma$H2A.X, green) and telomere immuno-fluorescence *in situ* hybridization (red) within alveolar compartments. Arrows indicate $\gamma$H2A.X foci co-localizing with telomeres (TAF) (scale bar, 5 $\mu$m), shown at higher magnification on the right (images are from maximum intensity projection). (**f**) Mean number of $\gamma$H2A.X foci (left) and percentage of cells containing at least one TAF (right) were determined through quantification of $Z$-stack images with at least 100 cells per sample ($\times 100$ images) (mean $\pm$ s.e.m.; control $n = 10$ (grey), IPF $n = 27$ (red); $t$-test *$P \leq 0.05$).

epithelial cells within fibrotic foci and honeycomb lung, and accretion of TAF in lung tissue of individuals with IPF.

Hypothesizing that senescent cells mediate IPF pathology via their secretome, we assessed differential expression of SASP components in human IPF transcriptome data sets, focusing on growth and matrix remodelling factors, which play essential roles in proliferation and tissue reorganization[26]. Pregnancy-associated plasma protein A (*PAPPA*), which mediates insulin-like growth factor (IGF) signalling by cleaving IGF-binding proteins to liberate IGF[27], was robustly upregulated, as were *IGFBP2* and *4*, although to a lesser magnitude (Fig. 1a). Expression of several matrix-remodelling proteins (*MMPs*) strongly increased with disease severity and in concordance with profibrotic factors, including collagen, type I, $\alpha 1$ (*COL1A1*) and vascular cell adhesion molecule 1 (*VCAM1*, Fig. 1a), the latter of which is a potent mediator of fibroblast proliferation[28].

**Bleomycin injury induces cell senescence.** We next established experimental systems to further interrogate the cellular identity and mechanisms by which senescent cells exert their effects. Aerosolized intratracheal instillation of the chemotherapeutic agent bleomycin induces lung fibrosis in mice and recapitulates critical features of human IPF[15]. Using this murine model, we fluorescence-activated cell sorted (FACS) whole lungs isolated 14 days post bleomycin or phosphate-buffered saline (PBS) exposure based on cell-surface marker presentation (Fig. 2a–d) and conducted gene expression profiling on populations of fibroblasts (PDGFR$\alpha^+$, EPCAM$^-$, CD31$^-$ and CD45$^-$) (Fig. 2b,e), epithelial cells (EPCAM$^+$, PDGFR$\alpha^-$, CD31$^-$ and CD45$^-$) (Fig. 2c,f) and endothelial cells (CD31$^+$, PDGFR$\alpha^-$, EPCAM$^-$ and CD45$^-$) (Fig. 2d,g). We observed significant upregulation of *p16* within both fibroblasts and epithelial cells but not endothelial cells (Fig. 2e–g). Transcript

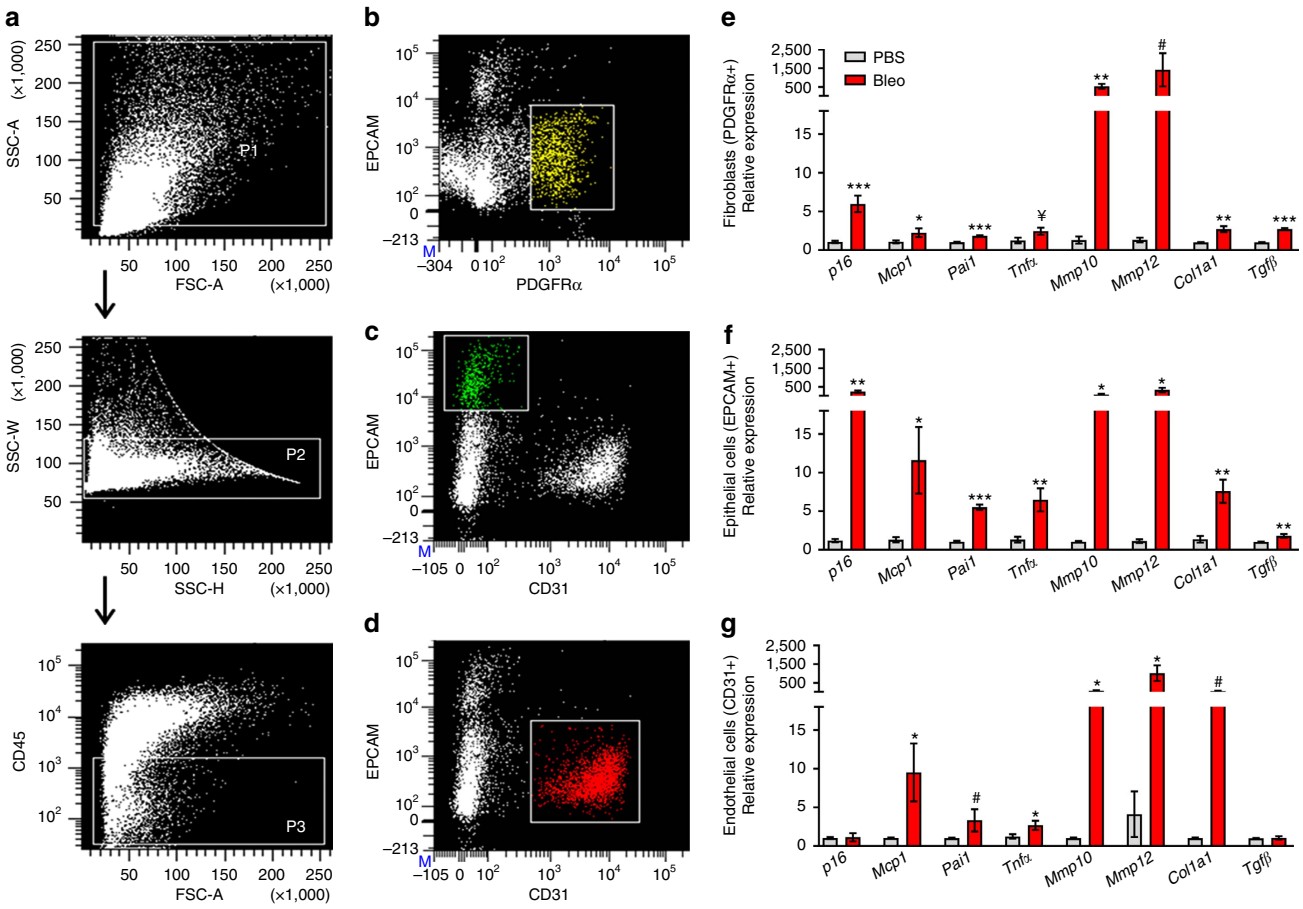

**Figure 2 | Bleomycin-induced senescence in murine lung cells.** (a–d) Gating strategy for isolation of fibroblasts, epithelial and endothelial cells from the lungs of mice 14 days post-aerosolized instillation of bleomycin (Bleo) or PBS. (a) Total single-cell suspensions (P1) were gated to exclude doublets (P2) and CD45 + cells (P3). (b) Fibroblasts (PDGFRα + , EPCAM − , CD31 − and CD45 − ), (c) epithelial cells (EPCAM + , PDGFRα − , CD31 − and CD45 − ) and (d) endothelial cells (CD31 + , PDGFRα − , EPCAM − , CD45 − ) were sorted from the P3 population. The expression of p16, SASP genes (Mcp1, Pai1, Tnfα, Mmp10, Mmp12) and fibrotic genes (Col1a1 and Tgfβ) were quantified by RT–PCR and are expressed relative to Hprt levels in sorted populations of (e) fibroblasts, (f) epithelial cells and (g) endothelial cells (mean ± s.e.m.; PBS n = 8 (grey), Bleo n = 6 (red); t-test, ***$P < 0.0005$, **$P < 0.005$, *$P < 0.05$, ¥$P ≤ 0.07$ and #$P ≤ 0.1$.).

levels of proinflammatory and profibrotic SASP factors, monocyte chemotactic protein 1 (Mcp1), plasminogen activator inhibitor 1 (Pai1), tumour necrosis factor-α (Tnfα), Mmp10, Mmp12, Col1a1 and transforming growth factor-β (Tgfβ) were also increased in fibroblasts (Fig. 2e) and epithelial cells (Fig. 2f). All three cell populations exhibited upregulation of Mmp10 and Mmp12 (Fig. 2e–g). Thus, similar to human IPF, bleomycin-mediated lung injury induces a senescent signature characterized by increased transcriptional activation of p16 and SASP components in fibroblasts and epithelial cells.

**The secretome of senescent fibroblasts is profibrotic.** To substantiate the SASP as a mediator of IPF pathology, we devised an in vitro assay to test fibrotic activation. Exposure of human IMR90 fibroblasts to 10 Gy irradiation induced senescence observable after 21 days, which was confirmed by staining for SA-β-gal (Fig. 3a) and expression profiling of senescence effectors (p16 and p53) and SASP factors (MCP1 and IL6) (Fig. 3b). Conditioned medium (CM) collected from senescent cells (SASP-CM), relative to sham-irradiated control cells (CCM), exhibited 3- to >200-fold increases in the abundance of established SASP proteins interleukin (IL)-1α, IL1β, IL6, IL10, MCP1, PAI1, VCAM1, MMP2, MMP12 and TGFβ (Fig. 3c).

To examine potential fibrogenic effects of the SASP, we treated naive IMR90 fibroblasts with non-CM (NCM), NCM + 2 ng ml⁻¹ TGFβ (positive control), CCM or SASP-CM and immunostained for α-smooth muscle actin (αSMA), an indicator of fibrotic activation[29]. SASP-CM induced αSMA signal intensity at a level comparable to TGFβ treatment (Fig. 3d). Sixty-two percent of fibroblasts treated with SASP-CM stained positive for αSMA protein, whereas only 21% of cells treated with CCM were αSMA positive. We next utilized traction force microscopy[30] to determine whether fibrotic activation translated to changes in contractile behaviour. SASP-CM-treated fibroblast exhibited significantly greater traction forces, relative to CCM-treated fibroblasts (Fig. 3e). IMR90 cells treated with SASP-CM also expressed higher levels of several fibrosis genes, including actin-α-2 (ACTA2, encoding αSMA), COL1A1, COL1A2 and fibronectin 1 (FN1), relative to NCM- and CCM-treated cells (Fig. 3f). Thus, our results demonstrate that the secretome of senescent fibroblasts robustly stimulates a fibrotic phenotype in healthy human fibroblasts. Importantly, CM collected from irradiated bronchiolar epithelial cells did not activate a fibrogenic response, as measured by αSMA fibrotic activation of naive IMR90 cells (Supplementary Fig. 2), suggesting that cell-type-specific SASP composition may differentially affect pathological phenotypes within the lung.

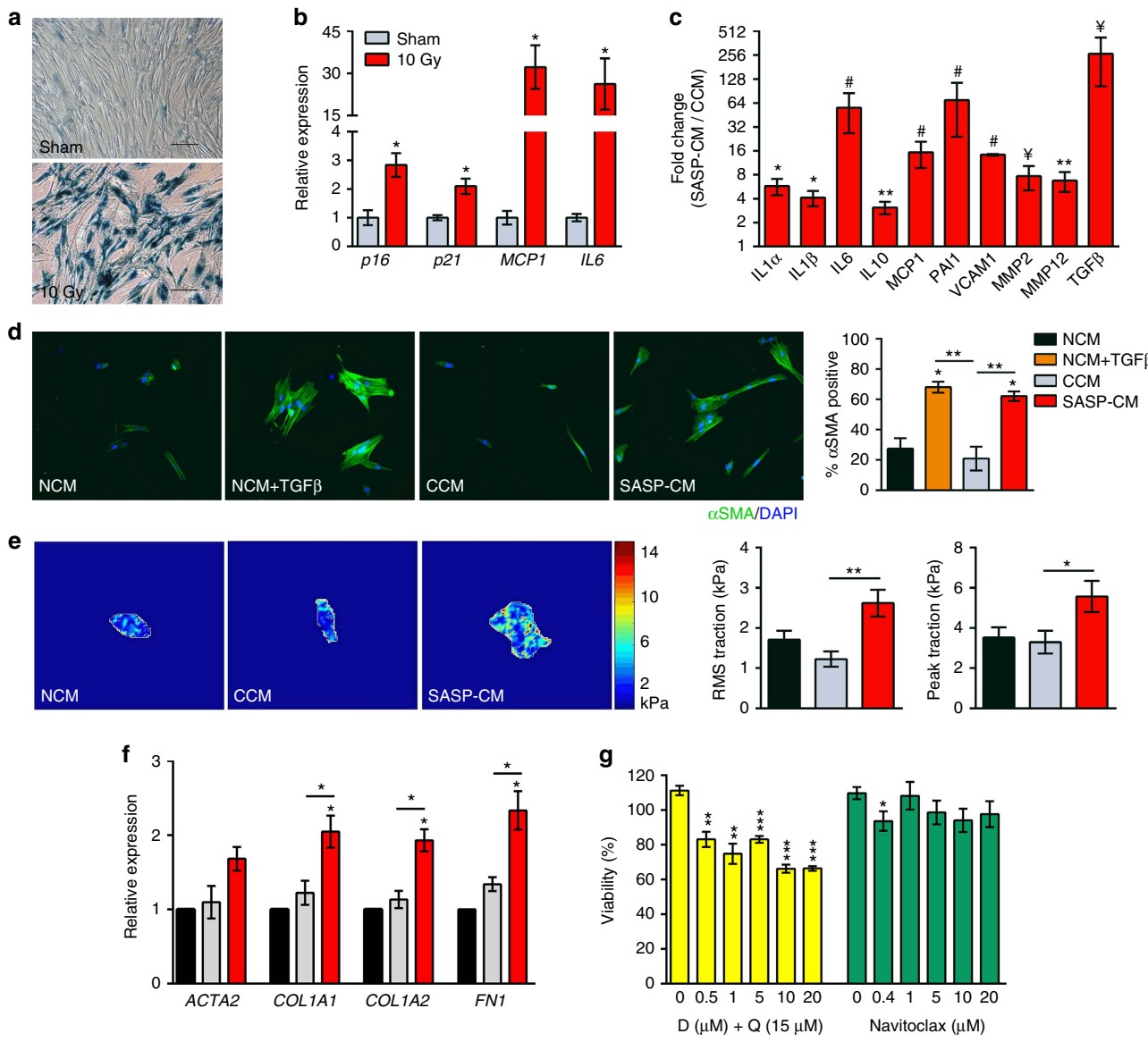

**Figure 3 | The secretome of senescent fibroblasts is profibrotic.** IMR90 lung fibroblasts were exposed to irradiation (10 Gy). Twenty-one days later, senescence was confirmed by (**a**) SA-β-gal staining (scale bar, 100 μm) and (**b**) RT–PCR assessment of *p16*, *p21*, *MCP1* and *IL6* expression relative to *TBP* levels in 10 Gy exposed (red) and sham-treated (grey) cells (mean ± s.e.m.; n = 3, *P ≤ 0.05), as well as (**c**) immunoanalysis of secreted SASP components within 10 Gy-exposed CM (SASP-CM), relative to CCM (mean ± s.e.m.; n = 4, **P < 0.01, *P ≤ 0.05, ¥P = 0.08 and #P ≤ 0.1.) (**d**) IMR90 cells were treated with media collected 21 days post 10 Gy, -sham exposure or control as follows: NCM (black), NCM + 2 ng ml⁻¹ TGFβ (orange), CCM (grey) or SASP-CM (red). IMR90 cells were treated with the indicated media for 72 h then immunostained for αSMA (green) and DAPI (blue). Percentage αSMA-positive cells were determined blindly, using a visual threshold (mean ± s.e.m.; n = 2–4 independent experiments; **P < 0.01 and *P ≤ 0.05). (**e**) IMR90 cells were plated onto 6.4 kPA matrices for traction force microscopy in the presence of the indicated media for 72 h. Representative traction heat maps, root mean square (RMS) traction and peak traction are depicted (mean ± s.e.m.; n = 2–4 independent experiments with a minimum of 20 independent cells per condition; **P < 0.01 and *P ≤ 0.05 versus control CM). (**f**) IMR90 cells were treated with indicated media for 72 h before RNA isolation. RT–PCR expression of *ACTA2*, *COL1A1*, *COL1A2* and *FN1* were measured relative to *GAPDH* levels (mean ± s.e.m.; n = 2–4 independent experiments, *P ≤ 0.05). (**g**) Human primary lung fibroblasts were exposed to irradiation (10 Gy). Twenty days post irradiation, cells were treated with the indicated concentrations of DQ (yellow) or navitoclax (green). Cell viability 3 days after drug treatment was measured by ATPLite assays and is indicated as a percentage of plating density at day 0 of treatment. (mean ± s.e.m.; n = 4, *t*-test, ***P < 0.001, **P < 0.01 and *P ≤ 0.05).

**Senescent fibroblasts are eliminated by DQ treatment.** Based on these findings, we reasoned that senescent cells might be viable pharmacological targets to counter fibrotic pulmonary disease. Therefore, we tested the efficacy of drugs previously shown to selectively eliminate senescent cells and spare quiescent and proliferating cells by comparing DQ, a tyrosine kinase inhibitor and flavonol combination[22], and navitoclax, a Bcl-2 family inhibitor[21]. In a dose-dependent manner, DQ killed irradiation-induced senescent primary human fibroblasts more efficiently than navitoclax *in vitro* (Fig. 3g). A single dose of 20 μM D + 15 μM Q eliminated >33% of senescent cells within 3 days (Fig. 3g). Similarly, in IMR90 fibroblasts made senescent by etoposide pretreatment, treatment with DQ robustly reduced senescence, as measured by SA-β-gal staining relative to control, whereas navitoclax treatment had no observable effect (Supplementary Fig. 3).

**Senescent cell clearance mitigates fibrotic lung disease.** Using the Ink-Attac mouse model, in which a drug, AP20187 (AP), induces apoptosis through dimerization of FKBP-fused caspase 8, we previously discovered that clearance of p16-expressing cells improves parameters of physical health and function in progeroid mice[31]. Attenuation of age-related pathologies and extension of lifespan in chronologically aged mice has been demonstrated using the same transgenic strategy[32–34], and improvements in lung structure and function in aged mice have recently been demonstrated using a similar suicide-gene strategy[20]. Here we applied the bleomycin fibrosis model to Ink-Attac mice, to test the hypothesis that senescent cell clearance attenuates pulmonary dysfunction. In parallel, we tested the efficacy of DQ treatment. Ink-Attac mice received a single dose of bleomycin or PBS through aerosolized intratracheal administration; bleomycin induced lung fibrosis over a 3-week time course (Fig. 4a). Compared with PBS, bleomycin increased whole lung p16 expression by threefold (Fig. 4b). Pulmonary transcript levels of putative SASP factors Mcp1, Il6, Mmp12, Col1a1 and Tgfβ also increased following bleomycin administration (Fig. 4c). Senescent cell clearance through AP-mediated suicide-gene activation or senolytic DQ treatment significantly reduced p16 transcriptional levels in the lungs and, concordantly, blunted increases in Mcp1, Il6, Mmp12 and Tgfβ (Fig. 4c).

We also assessed whether senescent cell clearance altered inflammatory aspects of bleomycin-induced lung injury through analysis of bronchoalveolar lavage (BAL) fluid at endpoint. Total BAL cell counts were significantly increased in vehicle-treated bleomycin-exposed mice, and treatment with AP or DQ attenuated this increase. Differential analysis revealed a similar trend for macrophages, lymphocytes and neutrophils (Supplementary Fig. 4a). Cytokine protein levels within the BAL fluid were highly variable; however, apparent increases in MCP1 (Supplementary Fig. 4b) and IL6 (Supplementary Fig. 4c) were diminished following AP or DQ treatment.

To determine whether the molecular and cellular phenotypes of lungs derived from mice subjected to AP and DQ treatment corresponded to functional improvements, we conducted non-invasive, whole-body plethysmography[35]. Bleomycin injury resulted in a twofold increase in enhanced pause (Penh), an indirect estimate of airway resistance. This effect was diminished by AP and DQ treatment (Fig. 4d). We employed the forced oscillation technique in anaesthetized, tracheostomized mice before necropsy as a direct measure of lung compliance[36]. Bleomycin reduced lung compliance in vehicle-treated mice by 40%. Both AP- and DQ treatment minimized this impairment, limiting bleomycin-induced reductions in lung compliance to 15% (Fig. 4e). Despite striking improvements in lung function following senescent cell clearance, quantified reductions in lung fibrosis histopathology did not reach statistical significance (Supplementary Fig. 5). Of note, the primarily airway-centred fibrosis was immature and reflective of acute lung injury, characterized by fibroblastic proliferation and mixed inflammation, rather than collagen deposition.

We next evaluated whether systemic health parameters were improved following AP or DQ treatment. Monitoring of body weight, an important indicator of pathology severity following bleomycin challenge[37], revealed that vehicle-treated mice lost an average of 3.6 g, relative to baseline. In comparison, both AP- and DQ-treated mice lost approximately 1.6 g of body weight (Fig. 4f). To assess exercise capacity, a measure that integrates multiple physiological systems and is used in people with lung disease to gauge intervention responses, we conducted a graded treadmill exercise test. Mice that received bleomycin and were treated with vehicle ran substantially shorter mean and maximal distances to exhaustion than all other groups. Compared with bleomycin-injured mice treated with vehicle, mice treated with AP and DQ ran, on average, >37% further to exhaustion (Fig. 4g).

**Senescent cell retention impairs fibrosis resolution.** Taken together, our in vivo results strongly support the hypothesis that elimination of senescent cells improves fibrotic pulmonary disease when treatment is initiated at the onset of pathogenesis. We next sought to test the clinically important question of whether senescent cell clearance provides comparable benefit when initiated at a timepoint when pathology has fully developed. Ink-Attac mice contained a strong senescent signature in lung fibroblasts and epithelial cells 2 weeks post-bleomycin challenge (Fig. 2), a timepoint of well-established fibrotic remodelling in the lung[15]. Accordingly, we administered a single dose of bleomycin or PBS through aerosolized intratracheal administration to Ink-Attac mice. Two weeks post challenge, we randomized bleomycin-exposed mice to AP, DQ or vehicle treatment groups based on change in body weight and non-invasively measured airway resistance (Penh). Mice were treated for 2 weeks and were killed 4 weeks following bleomycin or PBS exposure (Fig. 5a). Body weight monitoring revealed that, similar to our previous experiments (Fig. 4f), mice that received bleomycin lost substantial body mass, relative to mice that received PBS. Upon treatment initiation on day 14, mean body weight of mice treated with AP plateaued, while vehicle-treated, bleomycin-exposed counterparts continued to lose weight (Fig. 5a), suggestive of physical benefit afforded by senescent cell clearance, even when treatment is initiated in later-stage disease.

Spontaneous resolution is an inherent limitation of the bleomycin IPF model, with the rate and variability of bleomycin-induced pathology resolution dependent on complex factors, including genetic background and the microbiome[38,39]. Indeed, both endpoint compliance (Supplementary Fig. 6) and comparison of Penh airway resistance at endpoint (4 weeks post challenge) to measures taken at the treatment randomization timepoint (2 weeks post challenge) revealed appreciable, albeit variable, improvement in all bleomycin-exposed groups, including vehicle-treated mice, with AP-treated mice showing greatest improvement (Fig. 5b). Thus, in our hands, the 3–4-week-period following a single bleomycin exposure represents a pivotal window of natural resolution (Figs 4 and 5a,b).

Critically, impaired capacity to resolve fibrotic bleomycin injury has been associated with accumulation of apoptosis-resistant, senescent myofibroblasts[3]. As we were unable to disentangle senescence-dependent and -independent events governing resolution, we queried whether p16 expression, a measure of senescent burden, was associated with indicators of disease persistence. Impaired lung function, as measured by endpoint airway resistance (Penh) (Fig. 5c) and airway compliance (Fig. 5d), correlated significantly with p16 transcript abundance. This finding is consistent with associations drawn in humans, in which increased p16 expression correlated significantly with reduced FVC, pulmonary diffusion capacity and physical function (Supplementary Fig. 1). Similarly, in bleomycin-exposed mice, expression of the inflammatory and fibrogenic SASP factors Mcp1 (Fig. 5e), Pai1 (Fig. 5f), Mmp10 (Fig. 5g) and Col1a1 (Fig. 5h) robustly correlated with p16 transcriptional activation. These results further support the notion that the presence and activities of senescent cells may dictate resolution capacity following bleomycin-induced pathology.

## Discussion
Cumulatively, we demonstrate that cellular senescence contributes to fibrotic lung disease and can be targeted to improve

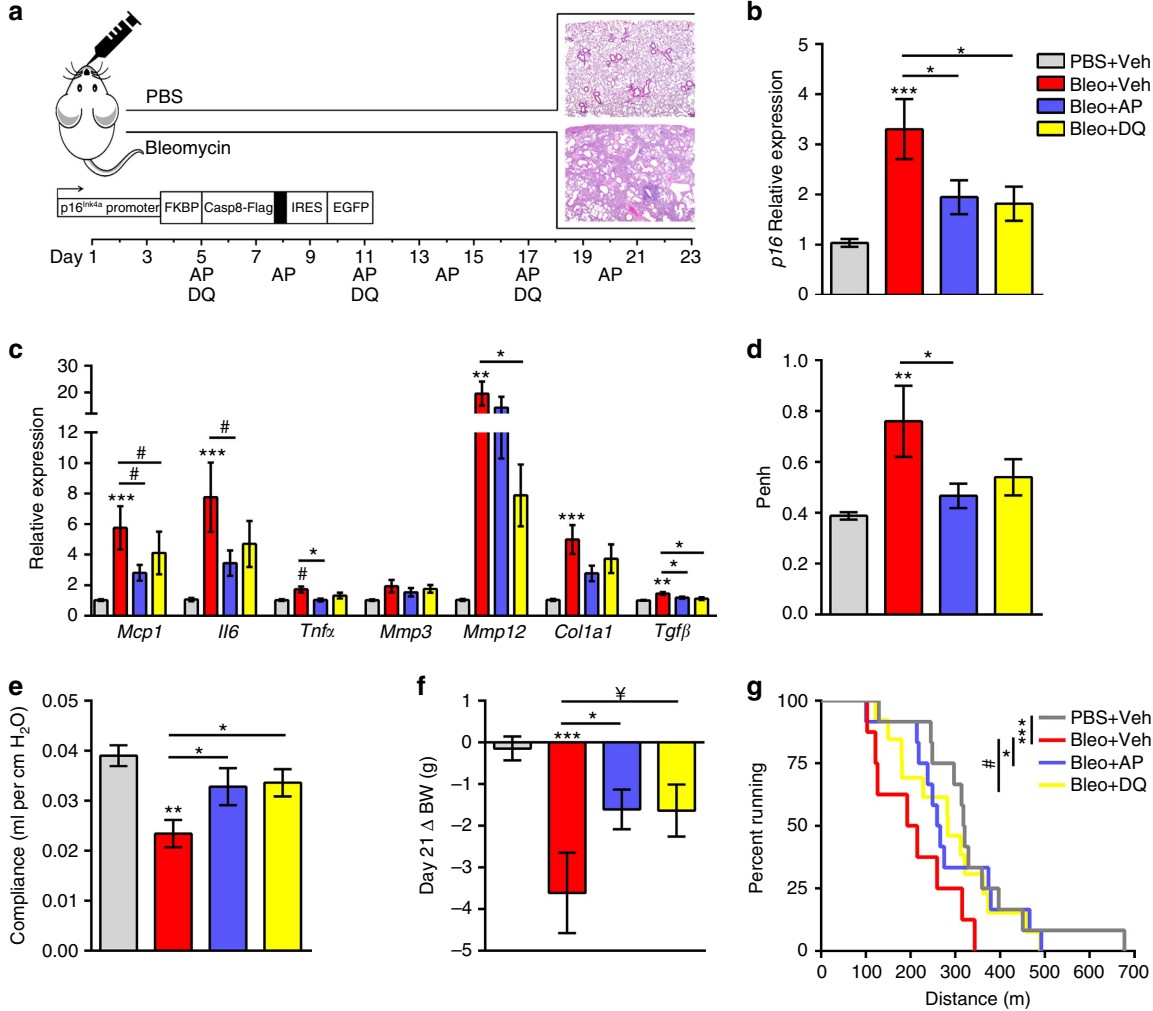

**Figure 4 | Senescent cell clearance improves pulmonary and physical health in bleomycin injury.** (**a**) Ink-Attac transgenic mice receiving bleomycin (Bleo) through aerosolized intratracheal instillation were randomized to receive vehicle (Veh), AP or DQ and compared with PBS-exposed mice treated with Veh. Treatment timeline is indicated. Mice were killed 3 weeks post challenge, a time point at which lung fibrosis peaks[15] (haematoxylin and eosin (H&E) panels top: PBS, bottom: Bleo). Lung expression of (**b**) *p16*, (**c**) SASP factors *Mcp1*, *Il6*, *Tnfα*, *Mmp3*, *Mmp12* and profibrotic factors *Col1a1* and *Tgfβ* were quantified by RT–PCR and are expressed relative to *Hprt* levels. (**d**) Whole-body plethysmography was used to assess enhanced pause (Penh), an indirect measure of airway resistance. (**e**) Lung compliance was ascertained by FlexiVent forced oscillation technique at endpoint. (**f**) Twenty-one-day body weight (BW) was compared with baseline body weight. (**g**) Exercise capacity was assessed through a treadmill test; distances ran to exhaustion are depicted (mean ± s.e.m.; PBS + Veh $n=13$ (grey), Bleo + Veh $n=8$ (red), Bleo + AP $n=12$ (blue), Bleo + DQ $n=13$ (yellow); linear regression model; \*\*\*$P<0.0005$, \*\*$P<0.005$, \*$P\le0.05$, ¥$P=0.08$ and #$P=0.1$).

pulmonary function and physical health. We show that senescence effectors and molecular markers of senescence-related DNA damage increase in the lungs of individuals with IPF (Fig. 1). This is associated with elevated transcription of SASP components. Cell-type analyses in the lungs of both human IPF (Fig. 1c,d) and bleomycin-injured mice (Fig. 2) demonstrate that fibroblasts and epithelial cells acquire senescent identities. Through *in vitro* testing we establish that the secretome of senescent fibroblasts may be a source of profibrotic signalling observed in pulmonary fibrosis (Fig. 3). We also discovered that senescent fibroblasts are efficiently killed by treatment with a senolytic cocktail, DQ (Fig. 3g and Supplementary Fig. 3).

Leveraging transgenic and pharmacological approaches, we show that bleomycin-mediated lung injury induces a molecular signature of senescence in mice. Elimination of senescent cells by suicide gene or senolytic strategies attenuates bleomycin-mediated impairments in lung function and physical health

(Fig. 4). Combined *in vitro* and *in vivo* results implicate the SASP as a mechanism by which senescent cells mediate fibrotic pulmonary pathology. Specifically, we show that the secretome of senescent fibroblasts contains an array of factors with established roles in regulating fibrotic and inflammatory aspects of IPF, including TGFβ, IL6 and MMP12 (Fig. 3c). Deletion of senescent cells reduces pulmonary expression of these factors (Fig. 4c). Senescent cells, therefore, are a viable origin of multiple signalling cascades that drive persistent fibroproliferative activation in IPF, and their clearance is a means to combinatorially down-regulate these processes.

Our murine results strongly support the hypothesis that senescent cell elimination potently influences health outcomes when animals are treated in early-stage pathogenesis, highlighting the potential utility of senolytics as an intervention strategy to be paired with early disease detection. Through demonstration that senescent cell clearance blunts expression of multiple profibrotic and proinflammatory factors, including *Mcp1*,

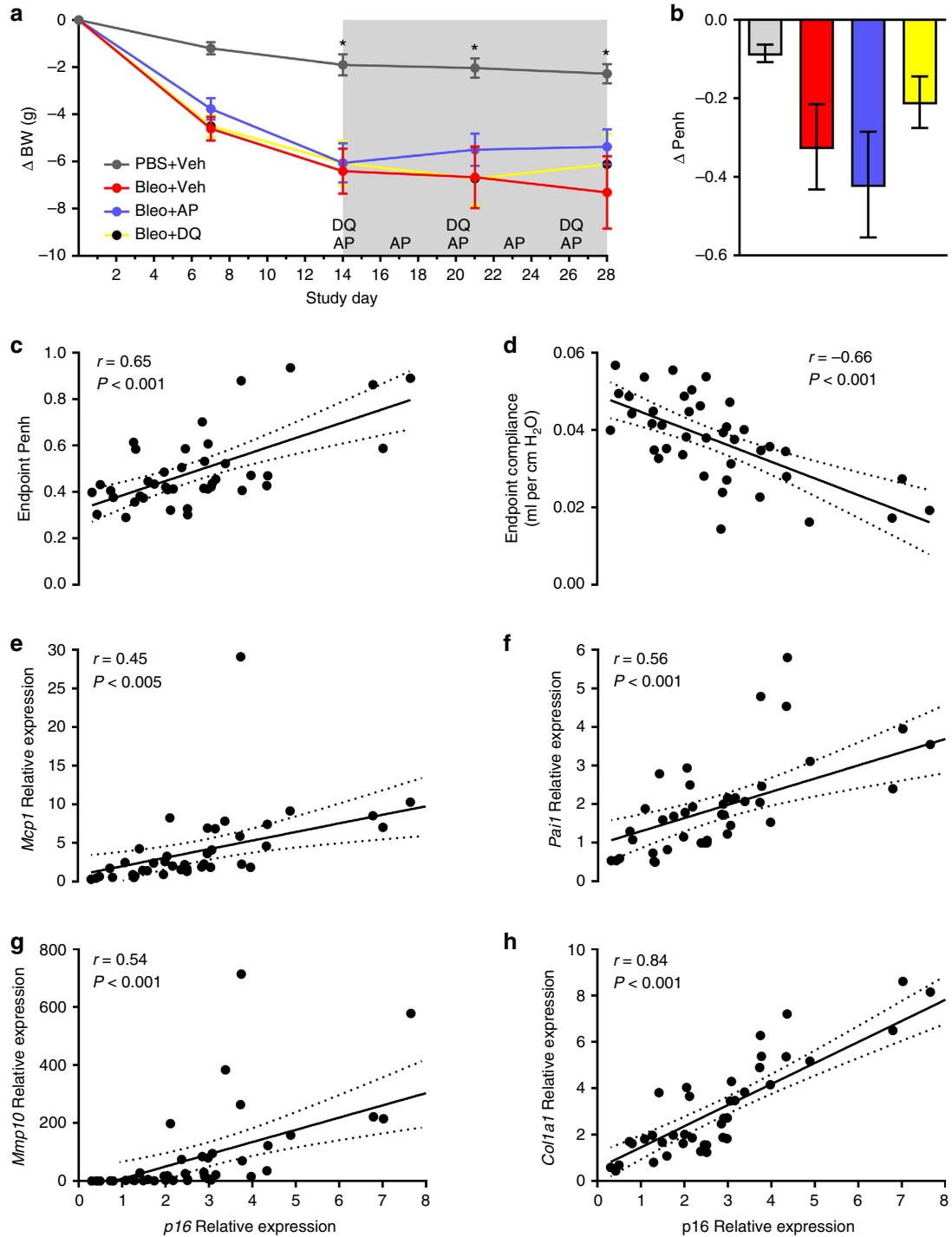

**Figure 5 | Retention of p16-positive cells impedes resolution of bleomycin lung injury. (a)** Ink-Attac transgenic mice receiving bleomycin (Bleo) through aerosolized intratracheal instillation were randomized to vehicle (Veh), AP or DQ and were compared with PBS-exposed, Veh-treated mice. Mice were treated as indicated from day 14–28 (grey shading). Body weight (BW) was monitored daily and is depicted as change in grams relative to baseline (*$P < 0.05$). **(b)** Endpoint (4 weeks post exposure) Penh levels were compared to Penh levels measured at treatment randomization (2 weeks post exposure). Lung *p16* expression levels measured by RT-PCR (normalized to *Hprt*) were compared with endpoint **(c)** Penh, **(d)** airway compliance, **(e)** *Mcp1*, **(f)** *Pai1*, **(g)** *Mmp10* and **(h)** *Col1a1* expression in bleomycin-injured mice (mean ± s.e.m.; PBS + Veh $n = 6$ (grey), Bleo + Veh $n = 13$ (red), Bleo + AP $n = 15$ (blue), Bleo + DQ $n = 14$ (yellow). Pearson's correlation statistics are indicated).

*Mmp12* and *Tgfβ*, our results support the premise that senescent cells are a source of persistent injury-response signalling underlying chronic activation in IPF. Signalling factor reductions coincided with striking improvements in pulmonary and physical function (Fig. 4c–g) in the absence of significant improvements in lung fibrosis (Supplementary Fig. 5). A lack of visible

histopathological fibrosis resolution may not reflect benefits at the cellular and subcellular level, and further, the short time course and heterogeneity intrinsic to the bleomycin model may limit observable fibrosis resolution. Indeed, future studies employing IPF models with extended disease courses that do not resolve, including transgenic manipulation,

 

irradiation or repetitive bleomycin administration, will be useful for conclusively determining the effects of senescent cells in later-stage disease[40,41]. We also provide evidence that senescent cell retention is associated with impaired ability to resolve bleomycin-induced injury. Taken together, our findings causally implicate senescence and the SASP in fibrotic lung disease, thereby revealing senescent cell elimination and SASP blockade as novel therapeutic approaches for the treatment of IPF.

## Methods

**Human samples and transcriptome data sets.** Microarray data, clinical parameters and lung tissue samples derived from control subjects and patients with IPF were obtained through the Lung Tissue Research Consortium (LTRC, (http://www.ltrcpublic.com)). The Mayo Clinic Institutional Review Board approved all aspects of the study using human data and biospecimens. Informed consent was obtained from all subjects who contributed medical data and biological specimens to the LTRC, comprising Mayo Clinic, University of Colorado, University of Michigan, Temple University and University of Pittsburgh.

IPF diagnosis was based on American Thoracic Society and European Respiratory Society criteria, and all IPF samples displayed typical patterns of usual interstitial pneumonia[42].

An independent set of LTRC-obtained tissues were applied to messenger RNA sequencing. RNA libraries were prepared from 200 ng of high-quality total RNA according to the manufacturer's instructions for the TruSeq RNA Sample Prep Kit v2 (Illumina, San Diego, CA). The concentration and size distribution of TruSeq libraries was determined on an Agilent Bioanalyzer DNA 1000 chip (Santa Clara, CA) and a final quantification, using Qubit fluorometry (Invitrogen, Carlsbad, CA), was done to confirm sample concentration. Libraries were loaded onto paired end flow cells at concentrations of 8–10 pM to generate cluster densities of 700,000 mm$^{-2}$ following Illumina's standard protocol using the Illumina cBot and cBot Paired end cluster kit version 3. The flow cells were sequenced as 51 × 2 paired end reads on an Illumina HiSeq 2000 using TruSeq SBS sequencing kit version 3 and SCS version 1.4.8 data collection software. Base calling was performed using Illumina's RTA version 1.12.4.2. RNAseq data and supporting information are publicly available. http://www.ncbi.nlm.nih.gov/geo/query/acc.cgi?acc=GSE92592.

**Mouse model.** All animal experiments were performed under protocols approved by the Mayo Clinic Institutional Animal Care and Use Committee (IACUC).

Anaesthetized male heterozygote Ink-Attac mice[32] on a C57BL/6 × BALB/c background aged 2.5–8 months old received PBS or 2 U kg$^{-1}$ bleomycin (Bleomycin for injection, USP, APP Pharmaceutical, LCC Schaumburg, IL) through aerosolized intratracheal delivery, as previously described[43].

For experiments employing senescent cell clearance as a means to prevent pulmonary disease, mice were randomized to AP (10 mg kg$^{-1}$, six treatments) delivered by intraperitoneal injection, dasatinib (5 mg kg$^{-1}$) plus quercetin (50 mg kg$^{-1}$) delivered by oral gavage (three treatments) or vehicle groups 5 days post-intratracheal instillation and were killed 3 weeks post-intratracheal instillation. For experiments employing senescent cell clearance as a means to treat pulmonary disease, mice were randomized to the same treatment groups 2 weeks post-intratracheal instillation and were killed 4 weeks post-intratracheal instillation. Body weight was monitored daily. Mice that failed to exhibit bleomycin-induced weight loss by 7 days were excluded from the study.

Physical function was characterized at endpoint by measuring running time and distance using a motorized treadmill (Columbus Instruments, Columbus, OH)[44]. Mice were acclimated to the treadmill for 3 consecutive days for 5 min starting at a speed of 5 m min$^{-1}$ for 2 min, then 7 m min$^{-1}$ for 2 min, followed by 9 m min$^{-1}$ for 1 min, at an incline grade of 5%. The next day, animals ran on the treadmill at an initial speed of 5 m min$^{-1}$ for 2 min and every subsequent 2 min the speed was increased by 2 m min$^{-1}$ until the mice were exhausted (5% grade). Exhaustion was defined as the inability of the mouse to remain on the treadmill despite an electrical shock stimulus and mechanical prodding. Running time and distance were recorded.

At endpoint, airway responsiveness was determined through whole-body plethysmography (Buxco Electronics, Sharon, CT) and FlexiVent forced oscillation technique (Scireq, Montreal, Canada)[45]. Briefly, for whole-body plethysmography, conscious mice were placed in a whole-body plethysmographic chamber (Buxco Electronics) with a volume of 800 cm$^3$. After a few minutes for stabilization, baseline respiratory waveform and enhanced pause (Penh, an indication of airway resistance in mice) were determined. For FlexiVent assessments, mice were anaesthetized with an intraperitoneal injection of ketamine/xylazine and placed in a supine position. A tracheostomy was performed and a blunt tip cannula as an endotracheal tube was inserted into the trachea. Mice were mechanically ventilated by the Flexivent system and physiological parameters (heart rate, pulmonary resistance, elastance and compliance) were recorded by a pressure transducer that was attached to the ventilator system. Subsequently, BAL fluid and lungs were collected. Total leukocyte counts in BAL fluids were determined using a haemocytometer after staining with Randolph's stain. For cell differential analysis, cytospin preparations from BAL fluids were stained with Wright–Giemsa and >200 cells were analysed by using a standard cell morphology criteria. BAL cytokines were measured using ProcartaPlex immunoassays (Affymetrix eBioscience, Vienna, Austria). Investigators conducting functional measures were blinded to treatment groups during data collection. Sample sizes were based on previously published experiments, in which statistical differences were identified.

**Fluorescence-activated cell sorting.** Fully anaesthetized bleomycin- and PBS-exposed mice were perfused with cold PBS 14 days post exposure and the lungs were immediately dissected. The lungs were finely minced with a razor blade in a 100 mm petri dish in 1 ml of cold DMEM medium containing 0.2 mg ml$^{-1}$ Liberase DL (Roche, Indianapolis, IN) and 100 U ml$^{-1}$ DNase I (Roche). The mixture was transferred to 15 ml tubes and incubated at 37 °C for 30 min in a water bath. Enzymatic digestion was inactivated by adding DMEM medium containing 10% fetal bovine serum. The cell suspension was passed once through a 40 μm cell strainer (Fisher, Waltham, MA) to remove multicellular debris. Cells were then centrifuged at 1,300 r.p.m. at 4 °C for 10 min, washed once in PBS and resuspended in 0.2 ml of FACS buffer (1% BSA, 0.5 μM EDTA pH 7.4 in PBS). The single cell suspension was then incubated with anti-CD45-PerCp-Cy5.5, anti-CD31-PE, anti-PDGFRα-APC and anti-EpCAM-BV421 antibodies (1:200) (Biolegend, San Diego, CA) for 20 min on ice. After incubation, cells were washed with ice-cold FACS buffer and resuspended in 1 ml of FACS buffer. FACS sorting was conducted using a BD FACS Aria II (BD Biosciences, San Jose, CA). FACS-sorted epithelial cells, endothelial cells and fibroblasts were collected in 1.5 ml Eppendorf tubes containing RLT lysis buffer (Qiagen, Hilden, Germany), which were subjected to mRNA extraction, complementary DNA synthesis and RT–PCR analysis.

**Cell culture.** For experiments employing radiation-induced senescence, IMR90 (passages 5–7; ATCC, catalogue number CCL-186) and primary human fibroblasts (passages 3–6) were exposed to 10 Gy radiation or sham conditions using a RS2000 X-Ray Irradiator (RAD Source Technologies, Suwanee, GA). Normal bronchial epithelial cells (passages 4–7, Lot 7F3000, Lonza, Walkersville, MD) were exposed to 5 Gy or sham conditions. Senescence was confirmed by expression profiling (p16, SASP factors) and/or SA-β-gal staining[32] 21 days post radiation. For SA-β-gal staining, cells were fixed for 10 min in 2% formaldehyde + 0.2% glutaraldehyde in PBS at room temperature. Cells were rinsed with PBS and incubated overnight in SA-β-gal staining solution containing the following: 1 mg ml$^{-1}$ X-gal, 40 mM citric acid/sodium phosphate buffer pH 6.0, 5 mM potassium ferrocyanide, 5 mM potassium ferricyanide, 150 mM sodium chloride and 2 mM magnesium chloride in water. Cells were then rinsed in PBS and stored at 4 °C until imaging. Media applied 20 days post irradiation (or sham) was collected 24 h later and was diluted with NCM to reflect equivalent source cell concentrations. ProcartaPlex immunoassays (Affymetrix eBioscience) were used for CM secretome composition assessment, according to the manufacturer's specifications. Non-irradiated IMR90 cells (passages 5–7) were treated with NCM, NCM + 2 ng ml$^{-1}$ TGFβ, CCM or SASP-CM and applied to downstream analyses as indicated. Primary human fibroblasts collected from human patients were authenticated through αSMA staining. For senolytic treatment, sham and irradiated primary human fibroblasts were treated with indicated concentrations of DQ or navitoclax for 3 days, starting at day 20 post irradiation or sham exposure. Cell viability after drug treatment was measured by ATPLite Kits (PerkinElmer; Waltham, MA) performed according the manufacturer's instructions. Luminescence was read using a multi-scan plate reader (Fisher).

For experiments utilizing etoposide-induced senescence, IMR90 human lung fibroblasts (passage 10; ATCC) were treated for 48 h with 20 μM etoposide (Enzo Life Sciences, Farmingdale, NY). Etoposide was removed and replaced with fresh media. Six days after etoposide removal, approximately 60% of IMR90 cells were SA-β-gal positive. Cells were treated for 48 h with 20 μM D + 15 μM Q or 10 μM navitoclax (or untreated control) and the percentage of SA-β-gal-positive cells was determined using a C12FDG-based senescence assay described as follows[46]. Briefly, etoposide-treated senescent IMR90 cells were seeded at a concentration of 5 × 10$^3$ cells per well in 96-well plates 24 h before treatment. Following addition of the drugs, the cells were incubated for 48 h under 20% O$_2$ conditions. For fluorescence analysis of SA-β-gal activity, cells were incubated for 1 h with 100 nM Bafilomycin A1 (Calbiochem, San Diego, CA). Next, 10 μM C$_{12}$FDG (Setareh Biotech, Eugene, OR) was added to the culture medium and the cells were incubated for 1.5–2 h. Ten minutes before analysis, the DNA intercalating Hoechst dye (2 μg ml$^{-1}$) was added to the cells. For quantitative analysis of cell number (Hoechst staining) and number of C$_{12}$FDG-positive senescent cells, a laser-based line scanning confocal imager IN Cell 6000 Analyzer was used. All samples were analysed in duplicate with three to five fields per well.

**Immunoanalysis and histopathology.** Formalin-fixed, paraffin-embedded (FFPE) tissue blocks corresponding to IPF (n = 27) and control (n = 10) subjects obtained from the LTRC were sectioned at 5 μm. For p16 immunohistochemistry, the FFPE unstained slides were deparaffinized through standard methods. Pretreatment consisted of CC1 Mild Cell conditioning (EDTA for 30 min) with Ventana Ultraview detection system (Ventana Medical Systems, Inc., Tucson, AZ) run on

 

Ventana XT Benchmark autostainer. The Ventana predilute CINtec p16 Histology (mouse monoclonal antibody clone E6H4) was used as the primary antibody (32 min at 37 °C). Appropriate positive and negative controls were performed. The expression of p16 was considered positive when both nuclear and cytoplasmic staining was identified.

TAF immuno-fluorescence *in situ* hybridization analysis was conducted as previously described[25]. Briefly, FFPE human lung tissue sections were de-waxed in Histoclear and hydrated in an alcohol gradient cascade. Antigen retrieval was achieved by boiling slides in 0.01 M citrate buffer. Cooled and washed slides were blocked in normal goat serum (1:60) in BSA/PBS and primary antibody (rabbit polyclonal anti-γH2AX 1:200, Cell Signaling, 9718) was applied and incubated at 4 °C overnight. Washed slides were then incubated with secondary antibody for 30 min (biotinylated goat anti-rabbit IgG, Vector Labs, BA-1000), washed three times in PBS and incubated with Avidin DCS (1:500) for 20 min. Slides were dehydrated in an alcohol gradient cascade and denatured for 5 min at 80 °C in hybridization buffer (70% formamide (Sigma-Aldrich, St Louis, MO), 25 mM MgCl$_2$, 1 M Tris pH 7.2, 5% blocking reagent (Roche)) containing 2.5 µg ml$^{-1}$ Cy-3-labelled telomere-specific (CCCTAA) peptide nuclei acid probe (Panagene), followed by hybridization for 2 h at room temperature in the dark and washing. Sections were incubated with 4,6-diamidino-2-phenylindole (DAPI), mounted and imaged. In-depth Z-stacking (a minimum of 40 optical slices with × 100 oil objective) followed by Huygens (SVI) deconvolution was employed for imaging.

For *in vitro* immunocytochemistry, IMR90 cells were plated into eight-well chamber slides (Thermo Fisher Scientific, Waltham, MA) and after attachment cells were treated for 72 h with prepared CM. Cells were fixed in 3.7% formalin (Sigma-Aldrich), permeabilized in 0.25% Triton X-100 (Sigma-Aldrich) and then blocked with 1% BSA for 1 h. Cells were incubated overnight with a fluorescein isothiocyanate-conjugated mouse monoclonal antibody against αSMA (F3777, Sigma-Aldrich) diluted 1:200 in PBS with 1% BSA. Cells were then washed and mounted with ProLong Antifade with DAPI (Thermo Fisher Scientific). Slides were imaged using a Cytation5 (BioTek, Winooski, VT) microscope at × 20 magnification. For scoring, an observer blinded to the treatment conditions counted αSMA-positive cells using a visual threshold for bright fibrous staining and the observer was blinded to the treatment.

For murine fibrosis assessment, FFPE lung tissue blocks were sectioned at 5 µm and subjected to haematoxylin and eosin and Masson's trichrome staining. Sections were reviewed by a blinded pathologist and approximately half of the specimens were scored by a second blinded pathologist to confirm agreement. Specimens were scored according to an eight-tier, modified Ashcroft scale[47].

**Traction force microscopy.** Traction analysis was conducted as previously described[31]. Briefly, polyacrylamide substrates with shear moduli of 6.4 kPa were prepared as previously described, and fluorescent sulfate-modified latex microspheres (0.2 µm, 505/515 ex/em) (FluoSpheres, Life Technologies) were conjugated to the gel surfaces after treatment with 1 mg ml$^{-1}$ of dopamine hydrochloride (Sigma-Aldrich) in 50 mM HEPES solution (pH 8.5). IMR90 cells were plated on the gels overnight and treated as indicated before traction force measurements. Images of gel surface-conjugated fluorescent beads were acquired for each cell before and after trypsinization using a Nikon ECLIPSE Ti microscope at × 10 magnification. Traction forces were estimated by measuring bead displacement fields and computing corresponding traction fields using TractionsForAll (http://www.mayo.edu/research/labs/tissue-repair-mechanobiology/software).

**Real-time PCR.** For transcriptional analysis on sorted mouse lungs and IMR90 cells treated with CM or NCM, total mRNA from each cell population was isolated using RNeasy mini kits (Qiagen) according to manufacturer's instructions. Isolated RNA was purity- and concentration-assessed by nanodrop. cDNA was synthesized using SuperScript II reverse transcriptase or SuperScript VILO (Invitrogen). IMR90 fibroblasts exposed to sham or 10 Gy radiation conditions and mouse lung tissue samples were subjected to Trizol-based RNA extraction, followed by nanodrop concentration and purity analysis, and cDNA synthesis through M-MLV reverse transcription (Invitrogen). RT–PCR was performed using FastStart Essential DNA Green Master (Roche) and analysed using a LightCycler 96 (Roche) or Applied Biosystems TaqMan Fast Advanced Mastermix (Thermo Fisher Scientific) analysed using an Applied Biosystems 7500 Real-Time PCR System (Thermo Fisher Scientific). RT–PCR primers used in this study are listed in Supplementary Table 3 (Integrated DNA Technologies, Coralville, IA).

**Statistical analyses.** GraphPad Prism 6.05 and R 3.2.0 were used for statistical analysis and generation of graphs. Data are expressed as the mean ± s.e.m. $P \leq 0.05$ was considered statistically significant. Unless otherwise indicated, the statistical method used for multiple comparisons was one-way analysis of variance with Tukey's *post-hoc* comparison. Binary variables were compared using *t*-tests. Pearson's correlation coefficients were used to summarize biomarker and functional data relationships. For human subject characteristics, continuous and categorical variables were compared using the analysis of variance F-test and the $\chi^2$-test, respectively.

For RNAseq, specimens were randomly assigned to assay processing, to balance batch preparation, flow cell and lane. The primary endpoint for gene expression was number of reads per gene, which were sequenced cDNA strands mapped back to the reference genome. The number of strands per region was used to evaluate the expression level of the region. Sample quality was evaluated using box plots to visualize gene counts per sample. In addition, minus versus average plots were used to assess global bias. The influence of GC content and gene length on gene expression was also examined. Transcripts quantified by RNAseq that had median counts of <32 in both control ($n = 19$) and IPF ($n = 20$) groups were considered 'non-expressed'. Normalized count data were evaluated in the same manner as the un-normalized count data, namely by utilizing minus versus average plots and visualizations of the GC content and gene length. Conditional quantile normalization using the cqn Bioconductor package[48] was applied to the RNAseq data, to reduce variability introduced by GC content, gene size and total gene counts per sample. Gene expression was evaluated using empirical Bayes estimates obtained through the use of edgeR in R. The results of the gene expression evaluation were ranked based on P-value and false discovery rate to account for multiple comparisons.

For human transcriptome analyses, we report expression changes that were identified in IPF versus control tissue by both microarray and RNAseq at a significance level of $q < 0.05$. For microarray comparisons, subject data were stratified based on FVC scores as follows: least severe (FVC >80%), $n = 17$; moderate (FVC 50–80%), $n = 60$; most severe (FVC <50%), $n = 16$; control, $n = 64$. Significance of differential expression was determined using linear regression via the functions lmFit and eBayes from the limma package[49]. Models were adjusted for age.

For mouse model data, linear regression was used to fit a model for each endpoint and included indicator variables adjusting for time period and group comparisons using Bleo-Vehicle as the comparison group. Model assumptions including identification of influential points and the distribution of the residuals were assessed and, when appropriate, transformations were used on the dependent variables to improve the model fit. The models were summarized using the Dunnett's test, which adjusts for multiple comparisons to a single control.

**Data availability.** The microarray data sets analysed in the current study are available in the National Center for Biotechnology Information Gene Expression Omnibus repository (https://www.ncbi.nlm.nih.gov/geo/query/acc.cgi?acc=GSE47460), under the accession code GSE47460. The RNAseq data sets generated in the current study are also available in the National Center for Biotechnology Information Gene Expression Omnibus repository (https://www.ncbi.nlm.nih.gov/geo/query/acc.cgi?acc=GSE92592), under the accession code GSE92592. The authors declare all other data supporting the findings of this study are available within the paper and its Supplementary Information files, and from the corresponding author on request.

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

## Acknowledgements
This publication was made possible by a Team Science Award from the Mayo Clinic CTSA grant number UL1TR000135 from the National Center for Advancing Translational Science (NCATS), a component of the National Institutes of Health (NIH) and a generous gift from the John E. and Virginia H. Kunkel Family (N.K.L.). This research was conducted while Marissa Schafer was a Glenn/AFAR Postdoctoral Fellow. We are grateful to the LTRC for collecting and providing subject specimens and sharing microarray data, and to Dr Robert Vassallo of Mayo Clinic for sharing mRNA sequencing data. We like to acknowledge the support of NIH R01AG53832 (N.K.L.), R01AG13925 (J.L.K.) and R01HL092961 (D.J.T.), the Glenn Foundation for Medical Research (J.L.K. and N.K.L.), the Connor Group, Noaber Foundation (J.L.K.), the Robert and Arlene Kogod Center on Aging, a David Phillips Fellowship funded by the Biotechnology and Biological Sciences Research Council (BBSRC) (BB/H022384/1) (J.F.P.) and a BBSRC grant (BB/K017314/1) (J.F.P.). We greatly appreciate the technical expertise and support of Glenda Evans, Kurt Johnson, Brian Kabat, Diane Grill, Ashley Brown and Pengyuan Zhang.

## Author contributions
This study was designed by M.J.S., T.A.W., K.I., D.J.T., H.K. and N.K.L. M.J.S., T.A.W., K.I., A.J.H., G.L., J.B., H.S., Y.Z., D.L.M., R.W.B., H.F., T.P. and M.C.A. collected data, which was analyzed and interpreted by M.J.S., T.A.W., A.J.H., G.L., E.J.A., A.L.O., J.B., M.C.A., J.F.P., D.J.T., H.K. and N.K.L. Additional expertise was contributed by T.T., Y.S.P. and N.K.L. The manuscript was drafted by M.J.S., T.A.W., K.I., A.J.H., G.L., E.J.A., J.B., H.S., J.F.P., D.J.T., H.K. and N.K.L. and revised by M.J.S., T.A.W., K.I., A.J.H., G.L., E.J.A., J.B., H.S., J.F.P., J.L.K., D.J.T., H.K. and N.K.L.

## Additional information

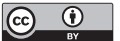

