## [Peer Review File · Nature Communications]

Reviewers' comments:

Reviewer #1 (expert in pulmonary fibrosis)

Remarks to the Author:

This is an important study that challenges existing paradigms on the role of cellular senescence in wound healing and fibrosis. Importantly, the new paradigm is well supported by the findings of senescent cells in fibroblastic foci in a human fibrotic disease, IPF (Fig. 1). While the cell model (Fig. 2) provides a plausible mechanism for how senescent cells may contribute to fibrotic progression and the in-vivo studies (Fig. 3) supports the beneficial roles of eliminating senescent cells, the specificity of the drug treatments is not clear.

Studies to show that the drug cocktail functions via elimination of senescent cells, preferably using a genetic approach, would strengthen the conclusions of this study.

Reviewer #2 (expert in cell senescence in mouse models)

Remarks to the Author:

This paper further explores the therapeutic benefits of eliminating senescent cells in diseases with abundant senescent cells, as it is the case of pulmonary fibrosis. This is a timely paper, correctly executed. However, I think that for a journal of high impact, such as Nature Communications, the authors should add further evidence.

Main request:

1. The therapeutic effect of senescence elimination is only shown with a genetic system (the INK-ATTAC mice). However there are now pharmacological interventions directed to the elimination of senescent cells and pioneered by the authors of the current paper, such as dasatinib/quercetin and navitoclax. In my opinion the paper would greatly improve its relevance if authors show that these pharmacological interventions also have therapeutic effect on pulmonary fibrosis.

Minor concerns:

2. The second sentence of the Introduction says that older organisms have a reduced ability to heal wounds and resolve fibrosis. A reference (or references) would be welcomed here.
3. Further down, it says that senescent cells paracrinely promote proliferation. Yes, I agree, but there are also many papers indicating that paracrinely promotes arrest and senescence.
4. The same sentence above also says that senescent cells paracrinely promote tissue deterioration. I am not aware of this. Please, add a reference or remove this part of the sentence.
5. Fig. 2g: I don't see the novelty of this figure relative to the paper of the authors on DQ in Aging Cell 2015.

Reviewer #3 (expert in lung fibrosis and senescence)

Remarks to the Author:

The manuscript on 'Cellular Senescence Drives Fibrotic Pulmonary Disease' by Schafer and colleagues used human tissues (IPF), primary human fibroblasts IMR90, mouse models, and chemical compounds to study the cellular senescence phenotype due to senescent fibroblasts drives pulmonary fibrosis. The role of senescence and senescence phenotype in human pulmonary fibrosis and mouse models (using bleomycin) has been shown earlier (Aoshiba et al Eur Respir J 2003; Aoshiba et al Expt Toxicol Pathol 2013; Stout-Delgado HW et al Am J Respir Cell Mol Biol 2016; and Yanai et al Aging 2015). However, the authors have used only male heterozygous Ink-Attac mice where up to ~25-30% progeria

phenotype in BubR1 KO (delayed senescence) has been shown by removal of p16 Ink expressing cells, along with chemical inhibitors to (dasatinib and quercetin) to inhibit senescence secretory phenotype in cells and mice.

Major comments:

1. Many key controls are missing in both cell culture and mouse experiments, e.g. TGF- β ; WT mice in bleomycin model with/without DQ and AP alone etc.
2. Differential cell counts (BAL), BAL cytokines, γ -H2AX foci and TAF in mouse lungs in WT and all four groups are needed.
3. It is not clear what cell type of the lung tissue shows increase in γ -H2AX foci and %TAF in controls vs. IPF lung tissue (Fig. 1e-f).
4. Representative better quality images for Fig. 2d. Need more cells in each field for Fig. 2d all the treatment groups.
5. SA- β -gal and p16 levels in all the treatments are needed for senescent and senescence phenotype.
6. Mouse survival data is not provided by various treatments including WT and INK-ATTAC mice.
7. Must show the classical parameters of fibrosis/scarring in vivo in all the groups along with lung staining (e.g. histology + fibrosis score, hydroxyl proline assay, etc).
8. Senescent cells are important for wound healing (Demaria et al Dev Cell 2014). Did the authors check whether the mice develop lung scarring during (impaired wound healing) or after the removal of p16 positive cells in mice and/or using AP or DQ or in combination.
9. Not clear why the drug AP was given every 2 days until day 20 and DQ was given every 5 days (3 times) until day 17.
10. Why not the compound DQ was given after the establishment of fibrosis i.e day 20 (therapeutic model of IPF). Percentage of senesced cells were eliminated by the drug (DQ) in bleomycin model.
11. Cell passage number and viability at early time points are needed
12. Some in vitro data on primary human lung fibroblast (normal and IPF) for SASP are needed.
13. Need to show D and Q alone data for treatments along with D+Q in several figures.
14. Patient/subject characteristics, details of samples used in RNA-seq and microarray analyses are missing.
15. SA- β -gal in IPF lungs, and cell type characteristics are not clear.
16. Some claiming sentences e.g. 'senescence cell elimination and SASP blockade as novel treatment of IPF and physical health/killing/suicide the cells etc' should be carefully revisited.
17. There seem to be a very little effect as both AP- and DQ-treatment minimized this impairment, limiting bleomycin induced reductions in lung compliance to 15%. Hence, the title may be changed.

Reviewers' comments:

Reviewer #1 (expert in pulmonary fibrosis)

Remarks to the Author:

This is an important study that challenges existing paradigms on the role of cellular senescence in wound healing and fibrosis. Importantly, the new paradigm is well supported by the findings of senescent cells in fibroblastic foci in a human fibrotic disease, IPF (Fig. 1). While the cell model (Fig. 2) provides a plausible mechanism for how senescent cells may contribute to fibrotic progression and the in-vivo studies (Fig. 3) supports the beneficial roles of eliminating senescent cells, the specificity of the drug treatments is not clear.

Studies to show that the drug cocktail functions via elimination of senescent cells, preferably using a genetic approach, would strengthen the conclusions of this study.

We appreciate the reviewer's positive comments on the strength of our data and the significance of our study. Our pharmacological approach was based on our previous work that showed dasatinib plus quercetin (DQ) eliminated senescent cells, but not quiescent cells, preadipocytes, MEFs, or HUVECs in culture [1]. In the current study, we compared the effects of DQ on primary human lung fibroblasts to another purported senolytic, navitoclax. Through additional experiments, our revised manuscript further suggests that DQ prevents the accumulation of SA-B-gal positive in IMR90 fibroblasts in culture (Supplementary Fig.2). Cell-based models do not perfectly replicate *in vivo* conditions. However, we put forward that the combination of our *in vitro* data with the notable similarities in the molecular and physiological phenotypes of bleomycin-exposed mice treated with DQ and AP20187, a drug that activates a suicide transgene in p16Ink4a expressing cells, provides a compelling argument that our drug cocktail targets senescent cells.

Reviewer #2 (expert in cell senescence in mouse models)

Remarks to the Author:

This paper further explores the therapeutic benefits of eliminating senescent cells in diseases with abundant senescent cells, as it is the case of pulmonary fibrosis. This is a timely paper, correctly executed. However, I think that for a journal of high impact, such as Nature Communications, the authors should add further evidence.

We thank the reviewer for the positive comments. We also appreciate the request for data to further support our conclusions. Correspondingly, our revised manuscript includes data from new analyses, *in vitro* studies, and two substantive *in vivo* experiments (new Fig. 2, Fig. 5). We feel the revised manuscript is comprehensive and well-supports the central hypothesis and our conclusions; cellular senescence is a mediator of fibrotic lung disease and may be novel therapeutic target to improve pulmonary and physical function.

Main request:

1. The therapeutic effect of senescence elimination is only shown with a genetic system (the INK-ATTAC mice). However there are now pharmacological interventions directed to the

elimination of senescent cells and pioneered by the authors of the current paper, such as dasatinib/querceetin and navitoclax. In my opinion the paper would greatly improve its relevance if authors show that these pharmacological interventions also have therapeutic effect on pulmonary fibrosis.

We agree with the reviewer that the use of pharmacological agents in the bleomycin model would be highly relevant. Thus, we employed parallel treatment of AP20187 (to activate a FKBP-CASP8 fusion protein suicide gene in p16-Ink-Attac mice) and a senolytic cocktail (DQ) in bleomycin-injured mice (Fig. 4). Our rationale for using DQ, rather than navitoclax, was based on our initial finding (now Fig. 3g) that DQ more effectively targets senescent fibroblasts *in vitro*. In the revised manuscript, we provide data from new complementary experiments that further support this observation (Supplemental Fig 2). In the mouse model, we show that both genetic and senolytic interventions improve lung compliance, exercise capacity, and body weight loss, indicative of significantly improved pulmonary and physical function (Fig. 4). Collectively, these data strongly support our premise that senescent cells may be novel and viable therapeutic targets for fibrotic lung diseases in humans

Minor concerns:

2. The second sentence of the Introduction says that older organisms have a reduced ability to heal wounds and resolve fibrosis. A reference (or references) would be welcomed here.

Age-associated impairments in wound healing have been strongly linked to alterations in the kinetics of repair processes, with both the timing and magnitude of inflammation, adhesion molecule expression, and matrix deposition differing in older organisms. We have added a reference [2] corresponding to these findings. Similarly, reduced lung fibrosis resolution as a function of aging is a central feature of IPF. We have added a reference from Hecker et al. [3], which provided early evidence that senescent myofibroblasts may play a key role in impaired fibrosis resolution in aged mice. Our new results (Fig.5), strongly support this premise.

3. Further down, it says that senescent cells paracrinely promote proliferation. Yes, I agree, but there are also many papers indicating that paracrinely promotes arrest and senescence.

We agree with the reviewer; the effects of senescent cells are highly context dependent. In the introduction, we highlight studies implicating senescence in both proliferative and anti-proliferative events, particularly in fibrotic contexts and/or the lung. Moreover, we state that the role of senescence in fibroproliferative pathogenesis is controversial. Thus, our findings significantly advance the field's understanding of cellular senescence in the context of a complex and devastating pathology.

4. The same sentence above also says that senescent cells paracrinely promote tissue deterioration. I am not aware of this. Please, add a reference or remove this part of the sentence.

Senescent cells secrete an array of active factors, including inflammatory molecules, MMPs, and ROS, that can disrupt neighboring cell function and structure, ultimately deteriorate tissue quality. The reference [4] provided is a review, which summarizes numerous studies in support of this notion.

5. Fig. 2g: I don't see the novelty of this figure relative to the paper of the authors on DQ in Aging Cell 2015.

Figure 2g (now 3g) is important in the context of our study. Previously, we demonstrated that DQ effectively killed senescent, but not quiescent, preadipocytes, MEFs, and HUVEC cells [1]. Navitoclax has since been identified as a senolytic in certain cell types [5]. Here, using cell viability (Fig. 3g) and SA-B-gal positivity (Supplemental Figure 2) as readouts, we demonstrate that DQ has far superior efficacy over navitoclax in senescent lung fibroblasts. These results, therefore, provided the rationale for selecting DQ over navitoclax as a treatment to test in the bleomycin model.

Reviewer #3 (expert in lung fibrosis and senescence)
Remarks to the Author:

The manuscript on 'Cellular Senescence Drives Fibrotic Pulmonary Disease' by Schafer and colleagues used human tissues (IPF), primary human fibroblasts IMR90, mouse models, and chemical compounds to study the cellular senescence phenotype due to senescent fibroblasts drives pulmonary fibrosis. The role of senescence and senescence phenotype in human pulmonary fibrosis and mouse models (using bleomycin) has been shown earlier (Aoshiba et al Eur Respir J 2003; Aoshiba et al Expt Toxicol Pathol 2013; Stout-Delgado HW et al Am J Respir Cell Mol Biol 2016; and Yanai et al Aging 2015). However, the authors have used only male heterozygous Ink-Attac mice where up to ~25-30% progeria phenotype in BubR1 KO (delayed senescence) has been shown by removal of p16 Ink expressing cells, along with chemical inhibitors to (dasatinib and quercetin) to inhibit senescence secretory phenotype in cells and mice.

Major comments:

1. Many key controls are missing in both cell culture and mouse experiments, e.g. TGF- β ; WT mice in bleomycin model with/without DQ and AP alone etc.

Cell Culture: In the revised manuscript, we have added TGF β treatment as a positive cell culture control (Fig.3D). As depicted, conditioned medium containing fibroblast SASP induces expression of alpha-SMA at a level comparable to induction by TGF β . Addition of this control, therefore, highlights the potent effect of the SASP, and we thank the reviewer for suggesting this addition.

We have also interrogated *Tgfb* expression in cell-type specific induction of senescence following bleomycin injury (New Fig.2e-g) and following senescent cell clearance (Fig.4c) and discovered that *Tgfb* is significantly upregulated in both fibroblasts and epithelial cells and significantly reduced following treatment with AP20187 or DQ. Thus, this potent profibrotic factor is a potential SASP component that may be reduced by senescent cell clearance. We feel that this finding enhances the relevance of senescent cell clearance as a therapeutic option for IPF.

Mouse Experiments: A recent study demonstrated that AP20187, which induces dimerization of the FKBP-CASP8 transgene to eliminate senescent cells in p16-Ink-Attac mice, has no

observable off-target effects in wild-type mice [6]. DQ is not targeted to the Ink-Attac transgene, but rather, selectively eliminates apoptosis-resistant senescent cells, an effect that has been established prior [1]. As control aspects of these agents have been investigated in the noted studies, we chose to dedicate additional animal experiments to addressing the questions of (1) which cell types acquire senescent phenotypes following bleomycin injury (New Fig. 2), and (2) whether senescent cell clearance is effective when initiated in later stage disease (New Fig. 5).

2. Differential cell counts (BAL), BAL cytokines, γ -H2AX foci and TAF in mouse lungs in WT and all four groups are needed.

We agree that bronchoalveolar lavage composition provides key information about pathology manifestation and severity. Thus, we have now analyzed total BAL cell counts. Furthermore, we report changes in select SASP cytokines within BAL fluid. As depicted in the new supplementary figure 3, bleomycin injury increased total immune cells, as well as individual populations of macrophages, lymphocytes, and neutrophils, within bronchoalveolar lavage fluid. Similarly, IL6 and MCP1 protein levels were higher in BAL following bleomycin exposure. Both genetic and pharmacologic strategies to target senescent cells mitigated these effects.

3. It is not clear what cell type of the lung tissue shows increase in γ -H2AX foci and %TAF in controls vs. IPF lung tissue (Fig. 1e-f).

We thank the reviewer for this comment. Cell-type-specific co-immunolabeling was not conducted throughout TAF analyses (γ -H2AX immunofluorescence + telomere immuno-FISH), although alveolar airways were the focused region of analysis. However, a pathologist with extensive expertise in clinical assessment of IPF reviewed human lung p16 staining and conclusively determined that both fibroblasts and epithelial cells are p16-positive in fibroblastic foci and honeycomb lung of IPF lung, while only rare epithelial cells are p16-positive in control lung (Fig. 1b-d).

To address the reviewer's concern and explore whether senescent cell type identity was conserved in the bleomycin mouse model of IPF, we FACS-sorted bleomycin-injured mouse lung and provide new data showing that both epithelial cells and fibroblasts express p16 (New Fig. 2e,f), as well as a compliment of SASP factors, while endothelial cells are p16-negative (New Fig. 2g). This addition enhances the understanding of senescence manifestation in fibrotic pulmonary disease, while further supporting bleomycin injury as a reasonable model of human IPF.

4. Representative better quality images for Fig. 2d. Need more cells in each field for Fig. 2d all the treatment groups.

Respectfully, based on the study team's expertise and experience with these methods, the images are of high-quality and representative of the indicated conditions. The depicted seeding density allows for accurate quantification. As requested, we have added TGF β treatment as a positive control image.

5. SA- β -gal and p16 levels in all the treatments are needed for senescent and senescence phenotype.

SA- β -gal staining must be conducted in fresh specimens, since β -galactosidase activity is significantly diminished or absent following fixation or freezing. Therefore, we are unable to conduct SA- β -gal staining on human samples or previously harvested mouse samples.

6. Mouse survival data is not provided by various treatments including WT and INK-ATTAC mice.

For the prevention study, mice were sacrificed three weeks post-exposure (Fig. 4), and for the treatment study, mice were sacrificed or four weeks post-exposure (Fig. 5). Separate survival studies were not conducted.

7. Must show the classical parameters of fibrosis/scarring in vivo in all the groups along with lung staining (e.g. histology + fibrosis score, , etc).

Per reviewer request, a pathologist who was blinded to treatment groups evaluated H&E and Masson's trichrome stained sections corresponding to mice described in figure 4 using a modified Ashcroft scale [7]. As indicated below, bleomycin-exposed mice displayed marked increase in pulmonary injury, relative to PBS-exposed counterparts. Mean lesion grades for AP20187- and DQ-treated mice were modestly diminished, but this reduction did not reach statistical significance. The improvements observed in lung function and physical health following treatment may not directly correlate with visible pathological improvements in this acute model of injury. Therefore, we did not include parameters of fibrosis or scarring in the manuscript.

8. Senescent cells are important for wound healing (Demaria et al Dev Cell 2014). Did the authors check whether the mice develop lung scarring during (impaired wound healing) or after the removal of p16 positive cells in mice and/or using AP or DQ or in combination.

We are aware of the study referenced by the reviewer that suggests senescent endothelial and mesenchymal cells and PDGF-AA, a component of the SASP, affect the kinetics of (but are not requisite for) cutaneous wound healing. It was noted during pathological grading (Q7) that bleomycin injury is characterized by high inflammation, cellular infiltration, and naïve fibrosis, but not significant collagen deposition observed in human IPF (i.e. scarring). Again, this is consistent with the acute nature of the bleomycin injury model. We are intrigued by the hypothesis that senescent cell activities and specific SASP components may be a source of persistent inflammation and fibrotic activation in IPF. Future studies will in part address what SASP factors that play either deleterious or beneficial roles in wound healing and fibrosis in the lungs. Although we are excited about the prospects of such experiments, they are beyond the scope of the current study.

9. Not clear why the drug AP was given every 2 days until day 20 and DQ was given every 5 days (3 times) until day 17.

Our treatment regimens were based on our and our collaborators' experience with these interventions. Additional studies to better understand optimal treatment regimens with these drugs and up and coming senolytics/senomorphics are warranted.

10. Why not the compound DQ was given after the establishment of fibrosis i.e day 20 (therapeutic model of IPF). Percentage of senesced cells were eliminated by the drug (DQ) in bleomycin model.

We agree that the influence of senescent cell clearance in later-stage, as compared to early-stage, pathology is an important experimental scenario to address. Based on this request, we conducted a comprehensive mouse experiment, (described in Fig. 5) to test whether treatment initiated in established disease (i.e., day 14) improved outcomes. As described in the revised manuscript, spontaneous, variable resolution is a limitation of the bleomycin injury model. Natural resolution by day 28 in this study limited our ability to decipher the “treatment” effects of our interventions (Fig. 5). However, this new experiment generated important and informative data. Specifically, and consistent with a recent study of chronological aging [3], resolution capacity is directly correlated with senescent cell persistence. Our data demonstrate that senescent cell burden, based on p16 transcript abundance, is positively associated with hallmarks of disease severity, including parameters of pulmonary function and the expression of inflammatory mediators, matrix remodeling proteins, and collagen (Fig. 5). In future studies, we plan to leverage complementary models of sustained pulmonary fibrosis, such as multiple bleomycin exposures, to better understand the extent to which interventions that target senescent cells can rescue persistent and progressive lung damage.

11. Cell passage number and viability at early time points are needed.

IMR90 fibroblasts were used at passage 5-7, and primary human fibroblasts were used at passage 3-6. This information has been added to the methods section.

12. Some in vitro data on primary human lung fibroblast (normal and IPF) for SASP are needed.

Based on this request, we conducted gene expression and proteomic analysis of conditioned media (CM) collected from primary human control and IPF lung fibroblasts. At baseline, no overt differences were noted in the expression of senescence or SASP biomarkers. SASP-containing CM was collected at baseline and 7, 14, and 21 days post 10 Gy irradiation. As depicted below, we didn't observe baseline differences in SASP factors among control and IPF fibroblasts. However, irradiation exposure induced a time-dependent increase in SASP factors in both control and IPF fibroblasts. No differences in the magnitude of these changes were noted between cells derived from normal and IPF lungs.

Within the human lung, senescent cells are relatively rare (Fig. 1b-d), and critically, they are cell-cycle arrested. Therefore, *ex vivo* cell populations are likely to contain few, or no, senescent cells. Accordingly, it is not surprising that baseline differences in the SASP were not observed. Since it has been previously shown that IPF cells have increased propensity to senesce *ex vivo* [8], we felt that further analyses in in this research trajectory would not add critical supportive results.

13. Need to show D and Q alone data for treatments along with D+Q in several figures.

We appreciate the reviewer's request; however, as noted in response to Reviewer #2, we have previously shown that D and Q are more effective in combination than in isolation. Respectfully, the requested experiments would not aid in the interpretation or impact of the manuscript.

14. Patient/subject characteristics, details of samples used in RNA-seq and microarray analyses are missing.

We thank the reviewer for calling this oversight to our attention. In the revised manuscript, we have provided 2 supplementary tables containing demographic data for subject samples applied to microarray (supplementary table 1) and RNAseq (supplementary table 2). We have also conducted additional statistical analyses of gene expression and phenotypic data. Our revised manuscript includes correlations between p16 expression and functional parameters, FVC, diffusion capacity, and physical function. This data is provided in supplementary figure 1 and complements our findings in the bleomycin mouse model (Q10).

15. SA- β -gal in IPF lungs, and cell type characteristics are not clear.

Please refer to questions 3 and 5.

16. Some claiming sentences e.g. 'senescence cell elimination and SASP blockade as novel treatment of IPF and physical health/killing/suicide the cells etc' should be carefully revisited.

We have reviewed these sentences and modified where appropriate.

17. There seem to be a very little effect as both AP- and DQ-treatment minimized this impairment, limiting bleomycin induced reductions in lung compliance to 15%. Hence, the title may be changed.

The title accurately reflects our collective data generated from three separate model systems (cells, mice, and humans).

- [1] Zhu, Y., Tchkonina, T., Pirtskhalava, T., Gower, A.C., Ding, H., Giorgadze, N., Palmer, A.K., Ikeno, Y., Hubbard, G.B., Lenburg, M., O'Hara, S.P., LaRusso, N.F., Miller, J.D., Roos, C.M., Verzosa, G.C., LeBrasseur, N.K., Wren, J.D., Farr, J.N., Khosla, S., Stout, M.B., McGowan, S.J., Fuhrmann-Stroissnigg, H., Gurkar, A.U., Zhao, J., Colangelo, D., Dorronsoro, A., Ling, Y.Y., Barghouthy, A.S., Navarro, D.C., Sano, T., Robbins, P.D., Niedernhofer, L.J. and Kirkland, J.L. (2015) The Achilles' heel of senescent cells: from transcriptome to senolytic drugs. *Aging Cell*.
- [2] Ashcroft, G.S., Horan, M.A. and Ferguson, M.W. (1998) Aging alters the inflammatory and endothelial cell adhesion molecule profiles during human cutaneous wound healing. *Laboratory investigation; a journal of technical methods and pathology* 78, 47-58.
- [3] Hecker, L., Logsdon, N.J., Kurundkar, D., Kurundkar, A., Bernard, K., Hock, T., Meldrum, E., Sanders, Y.Y. and Thannickal, V.J. (2014) Reversal of persistent fibrosis in aging by targeting Nox4-Nrf2 redox imbalance. *Science translational medicine* 6, 231ra47.
- [4] van Deursen, J.M. (2014) The role of senescent cells in ageing. *Nature* 509, 439-46.
- [5] Zhu, Y., Tchkonina, T., Fuhrmann-Stroissnigg, H., Dai, H.M., Ling, Y.Y., Stout, M.B., Pirtskhalava, T., Giorgadze, N., Johnson, K.O., Giles, C.B., Wren, J.D., Niedernhofer, L.J., Robbins, P.D. and Kirkland, J.L. (2016) Identification of a novel senolytic agent, navitoclax, targeting the Bcl-2 family of anti-apoptotic factors. *Aging cell* 15, 428-35.
- [6] Baker, D.J., Childs, B.G., Durik, M., Wijers, M.E., Sieben, C.J., Zhong, J., Saltness, R.A., Jeganathan, K.B., Verzosa, G.C., Pezeshki, A., Khazaie, K., Miller, J.D. and van Deursen, J.M. (2016) Naturally occurring p16(Ink4a)-positive cells shorten healthy lifespan. *Nature* 530, 184-9.
- [7] Hubner, R.H., Gitter, W., El Mokhtari, N.E., Mathiak, M., Both, M., Bolte, H., Freitag-Wolf, S. and Bewig, B. (2008) Standardized quantification of pulmonary fibrosis in histological samples. *BioTechniques* 44, 507-11, 514-7.
- [8] Yanai, H., Shteinberg, A., Porat, Z., Budovsky, A., Braiman, A., Ziesche, R. and Fraifeld, V.E. (2015) Cellular senescence-like features of lung fibroblasts derived from idiopathic pulmonary fibrosis patients. *Aging* 7, 664-72.

REVIEWERS' COMMENTS:

Reviewer #1 (Remarks to the Author):

My prior concerns regarding the specificity of pharmacologic agents have been satisfactorily addressed (using AP20187, a drug that activates a suicide transgene in p16Ink4a-expressing cells).

Reviewer #2 (Remarks to the Author):

I congratulate the authors for this nice and important work.

Reviewer #3 (Remarks to the Author):

NCOMMS-16-07902A

Ms title: Cellular Senescence Drives Fibrotic Pulmonary Disease

In this revised manuscript, the authors have included certain new data (Figures 2 and 5) by providing the therapeutic effect on removal of senescent cells using senolytic drugs. However, they failed to provide the conclusive proof that the removal of senesced cells is associated with protection against bleomycin-induced pulmonary fibrosis. The revised manuscript showed attenuation of lung function by pre-treatment (Fig. 4) of senolytic drugs (D =Tyrosine kinase inhibitor and Q =quercetin antioxidant), but unable to find the attenuation in post-treatment bleomycin-induced injury model (Supplemental Fig. 4) and SASP (Supplemental Fig. 3). Although they used the senescence reporter mice, the effect of bleomycin may be due to inhibition of signaling cascades in both the models (pre-treatment and post-treatment). Further, the authors failed to provide the pathological proof on attenuation of fibrosis in vivo in both the models.

The present study is well reflected as bleomycin-induced lung dysfunction and associated cellular and biochemical parameters are attenuated by pharmacological agents in mouse lung using the INK-ATTAC mice.

Minor:

There are several studies on linking cellular senescence with lung fibrosis, hence the authors should be careful in saying....

“Is known...we discovered elevated abundance of senescence effectors and senescence”

REVIEWERS' COMMENTS:

Reviewer #1 (Remarks to the Author):

My prior concerns regarding the specificity of pharmacologic agents have been satisfactorily addressed (using AP20187, a drug that activates a suicide transgene in p16Ink4a-expressing cells).

We thank the reviewer for her/his constructive feedback on our manuscript.

Reviewer #2 (Remarks to the Author):

I congratulate the authors for this nice and important work.

We thank the reviewer for their kind comment and appreciate her/his helpful suggestions.

Reviewer #3 (Remarks to the Author):

NCOMMS-16-07902A

Ms title: Cellular Senescence Drives Fibrotic Pulmonary Disease

In this revised manuscript, the authors have included certain new data (Figures 2 and 5) by providing the therapeutic effect on removal of senescent cells using senolytic drugs. However, they failed to provide the conclusive proof that the removal of senescent cells is associated with protection against bleomycin-induced pulmonary fibrosis. The revised manuscript showed attenuation of lung function by pre-treatment (Fig. 4) of senolytic drugs (D =Tyrosine kinase inhibitor and Q =quercetin antioxidant), but unable to find the attenuation in post-treatment bleomycin-induced injury model (Supplemental Fig. 4) and SASP (Supplemental Fig. 3). Although they used the senescence reporter mice, the effect of bleomycin may be due to inhibition of signaling cascades in both the models (pre-treatment and post-treatment). Further, the authors failed to provide the pathological proof on attenuation of fibrosis in vivo in both the models.

We appreciate the reviewer's remarks. We provide robust molecular and physiological evidence from mice that DQ phenocopies multiple effects of gene-mediated senescent cell clearance. Our cell-based data further show the direct effects of DQ on senescent lung fibroblasts; data that we provided in the original submission and complemented with additional assays/data from Dr. Paul Robbins laboratory at Scripps. Importantly, we also provide data that show senescent cell burden in mice and humans positively correlates with parameters of pulmonary fibrosis and function.

The short time course and heterogeneity intrinsic to the bleomycin model are widely recognized limitations. These shortcomings make pathological evidence of improvement with early-stage interventions and, even more so, late-stage interventions, challenging. A lack of visible histopathological fibrosis resolution may not reflect benefits at the cellular and subcellular level, and further, may limit observable fibrosis resolution. Indeed, future studies employing IPF models with extended disease courses that do not resolve,

including transgenic manipulation, irradiation, or repetitive bleomycin administration will be useful for conclusively determining the effects of senescent cells in later-stage disease. However, we demonstrate that DQ- and AP-mediated removal of senescent cells markedly reduces the pro-fibrotic molecular phenotype in bleomycin-exposed lungs. Importantly, these effects translate into clinically meaningful improvements in pulmonary and physical function. We also provide evidence that senescent cell retention is associated with impaired ability to resolve bleomycin-induced injury.

The present study is well reflected as bleomycin-induced lung dysfunction and associated cellular and biochemical parameters are attenuated by pharmacological agents in mouse lung using the INK-ATTAC mice.

Thank you. The parallels between the genetic and pharmacologic interventions on the outcomes of interest strongly support our conclusions that fibrotic lung disease is mediated, in part, by senescent cells, which can be targeted to improve health and function.

Minor:

There are several studies on linking cellular senescence with lung fibrosis, hence the authors should be careful in saying....

“Is known...we discovered elevated abundance of senescence effectors and senescence”

We appreciate the reviewer’s comment. In our revised manuscript, we have further emphasized the important contributions of previous studies to the body of evidence that cellular senescence is a plausible mediator of fibrotic pulmonary disease.